# The Art of *Barniz de Pasto* and Its Appropriation of Other Cultures

Yayoi Kawamura

Departamento de Historia del Arte y Musicología, Campus de Humanidades, University of Oviedo,
c/ Amparo Pedregal s/n, 33011 Oviedo, Asturias, Spain; kawamura@uniovi.es

**Abstract:** This study analyzes the techniques and decorative motifs of several works made using *barniz de Pasto*, highlighting their characteristics in order to establish comparisons with artistic phenomena of Asia and Europe. A possible link can be observed between *barniz de Pasto* and the Namban and Pictorial style Japanese export lacquer works of the 17th and 18th centuries. A search for similarity is justified by the documentary and material evidence of Japanese works created in these styles being transported from Japan to the Viceroyalty of New Spain by Manila galleons via the trade route between Acapulco and Callao. Additionally, traces of the Spanish culture have been recognized in *barniz de Pasto*. For example, printed images that circulated in the Viceroyalty of Peru have been observed on a coffer. This appropriation, also observed in the mural painting of a Central Andean church, and the presence of the image of Amaru, a Quechua deity, on the same coffer, marks the Central Andes as one of the possible places where the practice of *barniz de Pasto* could have been established. All of this points to Central and South America's great ability to appropriate foreign cultures and fuse them with their own during the viceregal period, as manifested in the art of *barniz de Pasto*.

**Keywords:** *barniz de Pasto*; Viceroyalty of Peru; lacquer; Japanese export lacquer; emblems; Amaru; appropriation

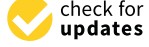



## 1. Introduction

*Barniz de Pasto* was an art form that was little known or studied, even among specialists in Spanish American art, up until two decades ago. This omission is gradually being corrected. In the case of Spain, the Museo de América in Madrid has been adding *barniz de Pasto* pieces to its collection and has even organized specific seminars attended by specialists from various countries to increase the spread of knowledge.[1] There have also been exhibitions that have contributed to a greater understanding of this art [1,2]. On the other hand, a growing interest in Spanish American art in the United States has bolstered the study of *barniz de Pasto*; exhibitions held at the Philadelphia Museum of Art, the Brooklyn Museum, and the Museum of Fine Arts, Boston, are especially noteworthy [3–5]. Likewise, studies on the Hispanic Society Museum and Library's works are of great interest [6,7]. Finally, material and botanical studies in the United States and Colombia have provided new insights about this art [8–16].

At present, it is commonly accepted that the production of *barniz de Pasto* took place in different areas of the Viceroyalty of Peru and is not exclusive to the area of Pasto, located in the southwest of Colombia, where this art is still practiced today. It is also believed that during this art form's evolution, several factors of diverse origins converged, not only European, but also Asian, thus correcting the traditional Eurocentric vision when analyzing Hispano-American art.

There are several recent studies that prove the arrival of Asian merchandise to the Viceroyalty of Peru beginning at the end of the 16th century, initially directly from Manila, and later from Acapulco. In this way, an important extension of the Manila galleon route was traced, linking Acapulco and the Port of Callao. This route was initiated and maintained by Mexican merchants and especially by the Peruvians, whose activities were supported by the abundance of silver produced in the Peruvian viceregal territory. The

transpacific merchandise trade operated with complete autonomy, outside the regulations dictated by the Spanish government's Council of the Indies [17–19]. In this way, luxury goods and, above all, textiles of Asian origin, were transported to the Viceroyalty of Peru, and the Peruvian silver was spent in Asia before it could reach Spain [20–22].

These historical and economic studies help us to understand that the traces of Asian art—especially lacquer, porcelain, and textile arts—are not only to be found in the Viceroyalty of New Spain, but also in the Viceroyalty of Peru. When analyzing lacquer art, the influence of Japanese export lacquer of the Namban (Figure 1) and Pictorial styles (Figure 2), has already been identified in Mexican lacquer art, or *maque*, and in the art of *enconchado*, for both painting and furniture. But we must now add *barniz de Pasto* to the conversation. Despite traditionally being called "barniz" in Spanish, or "varnish" in English, this art can be considered a variant of lacquer art. Furthermore, a great number of Chinese and Indian textiles could have influenced the dissemination of decorative elements or motifs with Asian roots in *barniz de Pasto* designs [23].

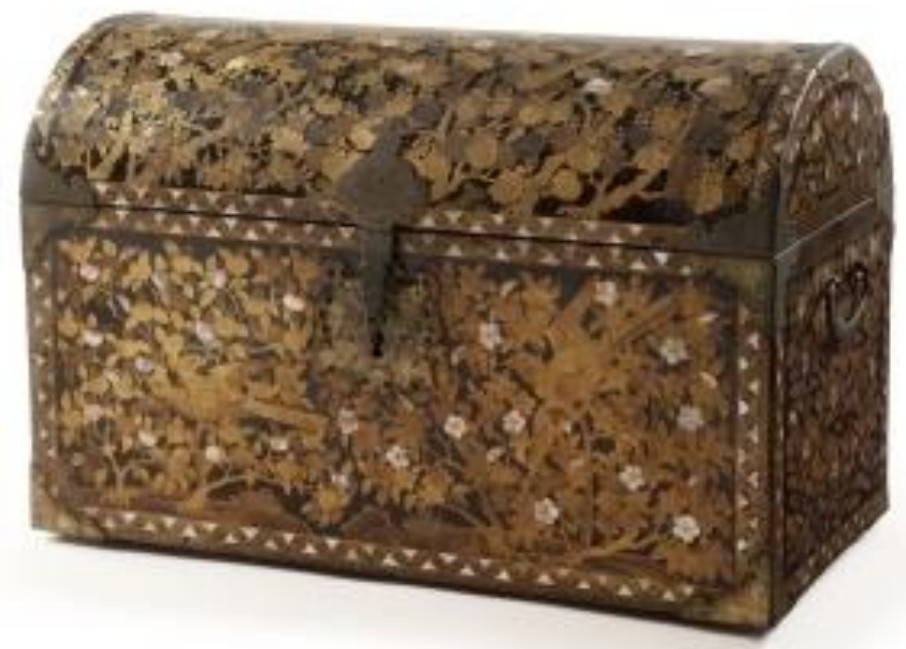

**Figure 1.** Namban style lacquer coffer, Parish of Cortes, Navarra, Spain. Japan, 1580–1630. Photography by R. Suárez Pascual.

Likewise, when studying *barniz de Pasto*, one must acknowledge the traces of the European world, since the dominant culture that was spreading in the Viceroyalty of Peru was that of Spain. Looking for concrete indications of the influences of this culture facilitates a greater understanding of the art of *barniz de Pasto*.

The present study applies the comparative method and analyzes the technical and iconographic aspects of several works of *barniz de Pasto*—including previously unstudied works—looking for parallels with other arts in order to point out possible influences and appropriations.

As the studies of this subject have been carried out and published hitherto by English-speaking and Spanish-speaking scholars, for a wide dissemination of this study, the Spanish version of this text is available as supplementary.

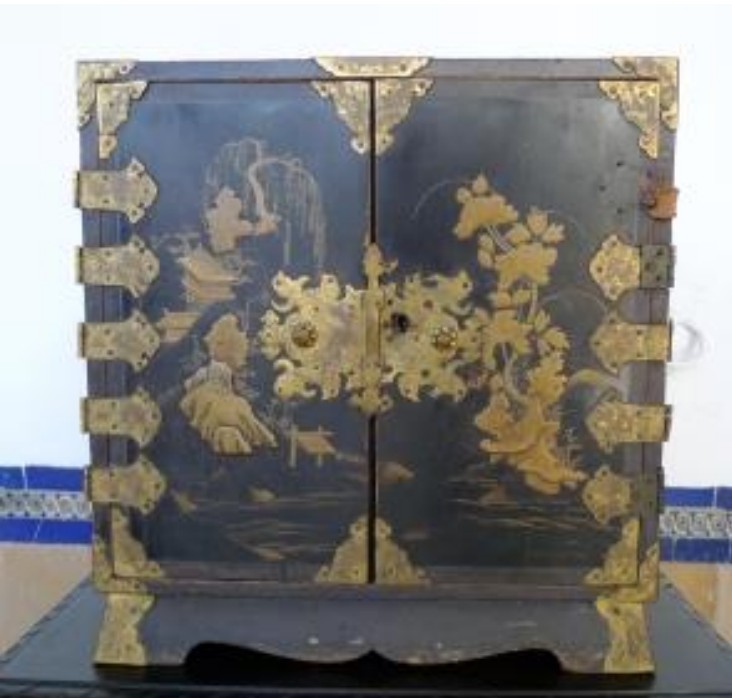

**Figure 2.** Pictorial style lacquer cabinet, Monastery of Santa Paula, Sevilla, Spain. Japan, 1640–1700. Photography by the author.

## 2. Asian Lacquer as a Source of Inspiration for *Barniz de Pasto*

In the late 16th century, during the Viceroyalty of Peru's consolidation, *barniz de Pasto* underwent a substantial change [15]. It ceased to be a filler material in the engraved grooves of various objects and began to be applied in the form of thin layers coating wooden surfaces, thus making objects more resistant and giving them a special shine. This shine gave rise to the term "barniz" in Spanish, or "varnish" in English. The technique of superimposition was developed for decorative purposes and involved cutting thin, differently colored layers into different shapes. This new way of working with *mopa-mopa* resin, i.e., in the form of thin layers [15] (pp. 55–57), has been considered a result of the hybridization of two cultures: the indigenous and the Spanish. The influence of painted Spanish imagery—polychrome wood sculpture—is especially noteworthy. The idea of inserting a thin sheet of gold or silver between the layers resulted in the glossy *barniz de Pasto.* This technique is also considered parallel to the gilding and *estofado* techniques utilized in Spanish imagery sculpture [13,15].

To better understand the unique technical and artistic evolution of *barniz de Pasto* from the last decades of the 16th century, another topic should be considered: the influence of foreign and even transpacific design elements on artistic innovation in the Americas. Apart from the impact that different decorative techniques from Spanish imagery may have had, the arrival of lacquered objects from the other side of the Pacific from the last quarter of the 16th century is an issue that should also be taken into account [1] (pp. 22–59) [24]. There is documentary and material evidence of shipments of lacquer of Asian origin, especially Japanese lacquer, from Manila to Acapulco. Japanese craftsmen created a specific genre, called the Namban (ca. 1580–1630) and Pictorial (ca. 1640–1700) styles, for export to the European world. Beginning at the end of the 16th century, Japanese lacquerware was highly appreciated by Spanish settlers in the Philippines. Antonio de Morga (Seville, 1559—Quito, 1636), judge, lieutenant to the governor-general, and captain general of the Philippines between 1595 and 1603, and author of *Sucesos de las Islas Filipinas*, made mention of the goods that arrived from the port of Nagasaki, among which lacquered objects were found [25] (pp. 313–314). Governor Rodrigo de Vivero (Mexico, 1564–Orizaba, 1636) made reference in his book *Relación de Japón* to the luxury Japanese goods shipped

to New Spain, which included lacquered cabinets [26] (p. 99). In 1612, the judge of the *Audiencia Real* of Manila, Juan Manuel de la Vega, sent a Japanese lacquer cabinet to Juan Ruiz de Contreras, secretary of the *Real y Supremo Consejo de Indias* in Madrid [1] (pp. 38–39). Philippine Governor Alonso Fajardo owned 19 pieces of Japanese lacquerware in 1624 [27]. These examples are evidence of the appreciation for Japanese lacquerware and its arrival via the Manila galleons to Acapulco, from where it was distributed to American lands and Spain [1] (pp. 53–59) [28].

There has not yet been an exhaustive study of the inventories of goods from the Viceroyalty of Peru that demonstrates the presence of Japanese lacquer works. However, a frame made with fragments of Namban-style lacquer from a private collection in Lima [29] (p. 15) and the Brooklyn Museum's Namban lacquer coffer, acquired in Peru in 1941 by Herbert J. Spinden [4] (p. 21), are proof of the arrival of Japanese lacquer in the Viceroyalty of Peru.

There are several studies on the similarities between *barniz de Pasto* and Japanese lacquer and that of Ryukyu, now known as the island of Okinawa, Japan. Codding points out the great similarity between the monkey-and-passion-flower-vine motif of the Hispanic Society Museum and Library's portable writing desk and the squirrel-and-grapevine motif of the lacquered stationery boxes from Okinawa [7] (p. 81). Kawamura also draws a number of parallels between *barniz de Pasto* and Japanese lacquer [30,31].

In terms of manufactured objects, there is an abundance of coffers, cabinets, portable desks, (Figure 3) and folding lecterns (Figure 4), which are also very common types of objects in Japanese Namban style lacquerware. The folding lectern form did not exist in Europe and was developed in Japan—perhaps inspired by lecterns from the Islamic world—because it was a form that took up little space as cargo on ships. Therefore, the existence of the folding lectern coated with *barniz de Pasto* is almost certainly inspired by the Namban lacquer lecterns (Figure 5), which were produced in significant numbers. On the other hand, in terms of unusual pieces, we can point to pineapple-shaped cups (Figure 6) and shaving basins (Figure 7), whose forms clearly come from silverware of European origin.

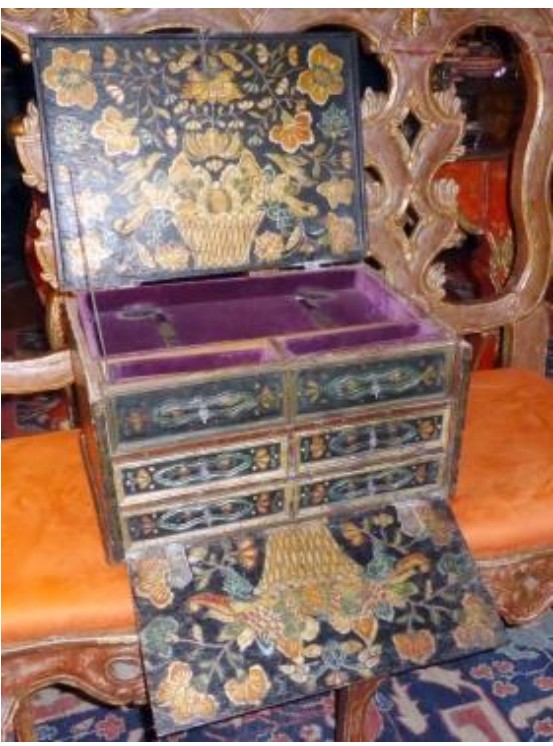

**Figure 3.** Barniz de Pasto cabinet, collection of Rodrigo Rivero Lake Int. Viceroyalty of Peru, 17–18th century. Photography by the author.

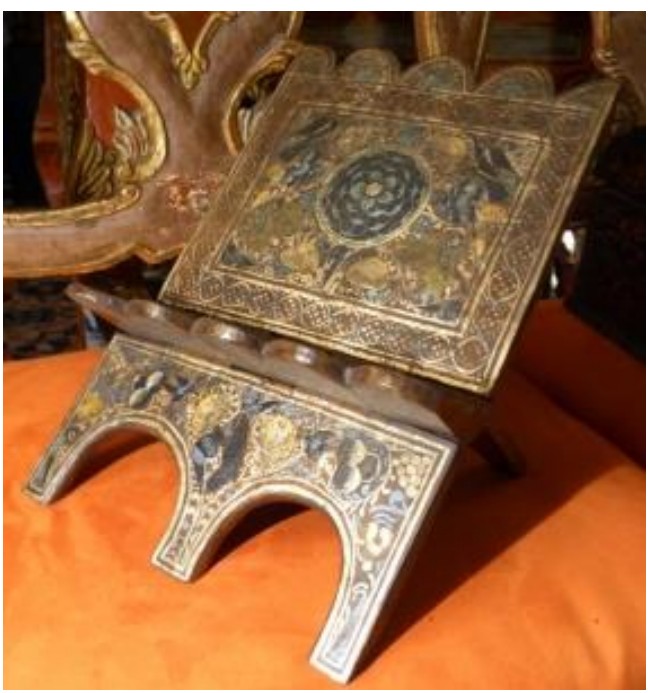

**Figure 4.** Barniz de Pasto folding lectern, collection of Rodrigo Rivero Lake Int. Viceroyalty of Peru, 17–18th century. Photography by the author.

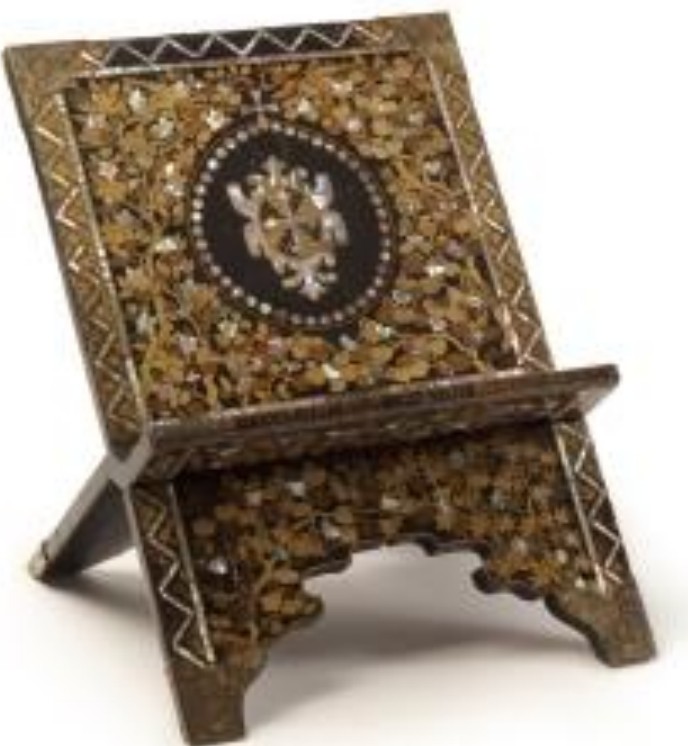

**Figure 5.** Namban style folding lectern, Monastery of San Esteban, Salamanca, Spain. Japan, 1580–1630. Photography by Rafael Suárez Pascual.

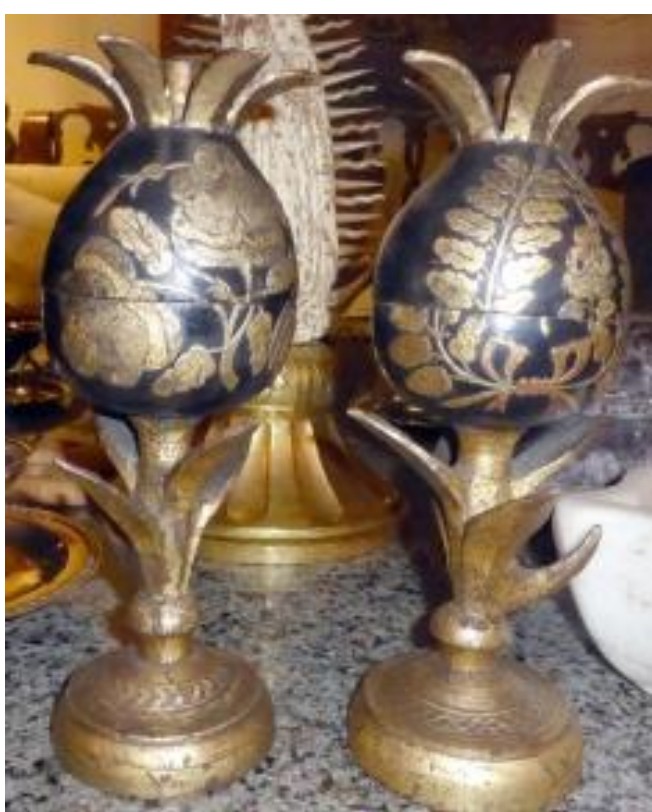

**Figure 6.** *Barniz de Pasto* cups, collection of Rodrigo Rivero Lake Int. Viceroyalty of Peru, 17–18th century. Photography by the author.

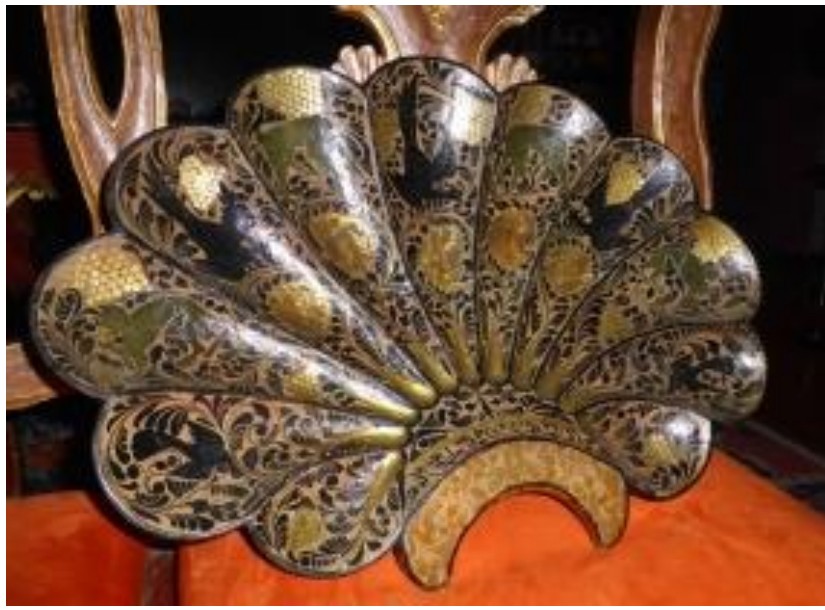

**Figure 7.** *Barniz de Pasto* shaving basins, collection of Rodrigo Rivero Lake Int. Viceroyalty of Peru, 17–18th century. Photography by the author.

The similarity between the decorative motifs of *barniz de Pasto* and Japanese lacquer objects is very evident. Namban lacquer is decorated with plants and trees, accompanied by birds and sometimes lions, and the presence of plants and birds is constant *in barniz de Pasto*. The manner of applying these decorative elements is also shared by the two arts. The surface is abundantly filled with decorative finishes in the manner of *horror* vacui.[2] Moreover, the

shine that silver leaf gives *barniz de Pasto* is reminiscent of the shine from gold dust—a technique of flat *makie* or *hira makie*—(Figure 8) and of mother-of-pearl inlay—a technique called *raden*—(Figure 9) of Namban lacquer.

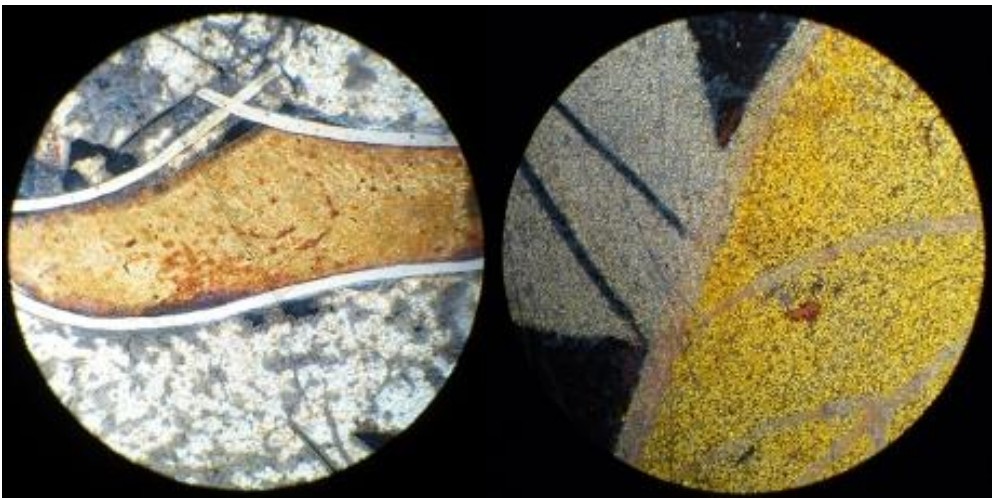

**Figure 8.** Silver sheet of *barniz de Pasto* coffer detail, private collection (**left**); and gold and silver powder of Namban style lacquer cabinet detail, Royal Monastery of La Encarnación, Madrid, Spain (**right**). Photography by the author.

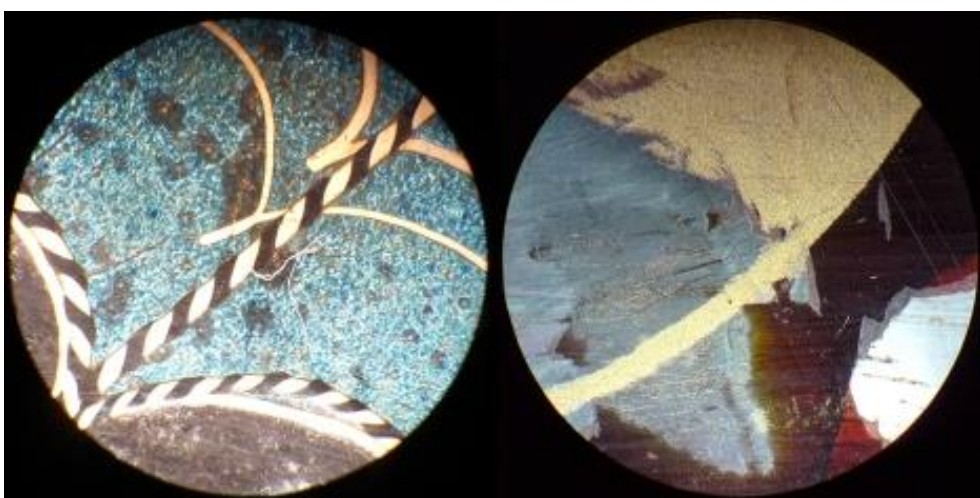

**Figure 9.** Silver sheet covered with a blue sheet of *barniz de Pasto* detail, private collection (**left**); and mother-of-pearl detail on a Namban style lacquer cabinet, Convent of Las Descalzas Reales, Madrid, Spain (**right**).

Another similarity with Japanese lacquer is the application of silver leaf cut in tiny squares or diamonds to express decorative details, as seen in the coffer of the parish of Mendigorría (Figure 10) or on a cabinet at the Museo de América in Madrid (inventory number: 01/08/2015). The same decorative resource is used in Japanese lacquer from the 13th century, the so-called *kirikane*, and a very similar decorative technique has been applied to Japanese paper since the 12th century. This type of Japanese paper was known in the Hispanic and European world. For example, the letters from Date Masamune addressed to the city of Seville and to Pope Paul V delivered by Ambassador Hasekura in 1614 and 1615, respectively, are written on gilded papers decorated with small squares of gold leaf.[3]

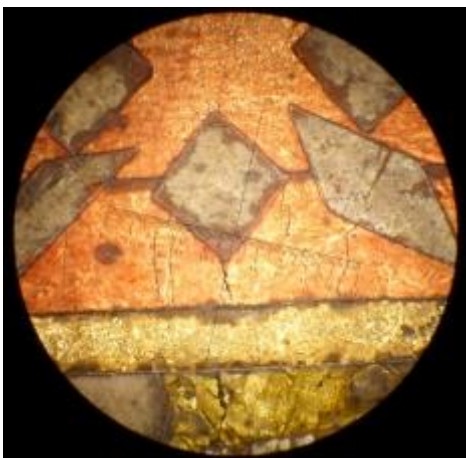

**Figure 10.** *Barniz de Pasto* coffer detail, Parish of Mendigorría, Navarra, Spain. Photography by the author.

The presence of bands acting as borders is another similarity observed in both arts, and variants of the repetitive geometric motifs observed in Namban lacquer can even be detected in *barniz de Pasto*. The border on the coffer from Pamplona Cathedral consists of linked diamonds reminiscent of the decorative pattern of the Japanese tradition called *shippô* (Figure 11), which is used in Namban lacquer. Additionally, the band of triangles running around the inside of the mouth of the gourd from The Hispanic Society Museum and Library (LS2400) is reminiscent of *kyoshimon*, a saw-tooth motif also used in Namban lacquer.

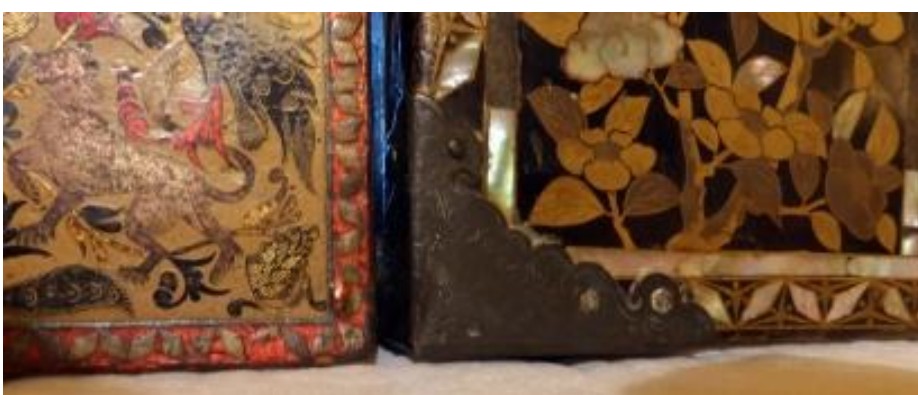

**Figure 11.** *Barniz de Pasto* coffer detail, Pamplona Cathedral, Navarra, Spain (**left**); and Namban style lacquer coffer, Parish of Cortes, Navarra, Spain (**right**). Photography by the author.

Of additional note is the abundant production of glossy *barniz de Pasto* objects with a black background, which brings to mind the chromatic combination of Japanese lacquer. Even when decorative motifs are applied in relief on a black background, as in the case of a pair of pineapple-shaped cups from a private collection (Figure 6), the appearance closely resembles the raised *makie* technique *(taka makie)*, a decoration in bold relief, applied on Pictorial style export lacquer (Figures 2 and 12).

It is true that, after the closure of Japan's borders in 1639, the only Europeans who could maintain trade, which was tightly controlled and limited by the Japanese government, were the Dutch. For this reason, it has traditionally been argued that the Namban style was replaced by the Pictorial style adapting to the preferences of the Dutch and that they were the only ones who transported the Pictorial style lacquers to Europe. However, there is evidence of the presence of two Pictorial style cabinets in the Museo Regional Casa de Alfeñique, in Puebla, Mexico (Figure 12), from the collection of José Luis Bello y González (1822–1907). There is still no definite proof that they were always in Mexico, but this is a very likely possibility [32]. There is evidence of other Pictorial style cabinets in the Monastery of Espíritu Santo and in the Convent of Santa Paula (Figure 2), both in Seville,

the only port for the arrival of ships from New Spain. On the other hand, a historical document dated 1719 speaks of the shipment of some formal gifts by the King of Siam to the King of Spain through the Governor of the Philippines. These gifts included two Japanese lacquer cabinets in the Pictorial style, along with pieces of Japanese porcelain.[4] The text speaks of "a large cabinet, of very fine Japanese *maque*". The accompanying drawing (Figure 13) was made to explain the assembly of a supplementary table, and for this reason, they did not draw the decorative details of the lacquered cabinet. However, because of its two doors with side handles, its showy keyhole plate, and large hinges, the piece of furniture is consistent with a Japanese cabinet in the Pictorial style. Everything indicates that even after 1639, luxury goods from Japan, dominated by export lacquer, gradually reached the Hispanic world via Chinese ships that had permission to dock at the Port of Nagasaki. These ships distributed the goods to different destinations, such as Siam or Manila, among other locales. Japan also maintained contacts with Korea through the sovereignty of Tsushima, and the Ryukyu Kingdom, although subjugated by the Japanese government, operated with some freedom in the South China Sea. In other words, there were routes other than the Dutch route for Japanese lacquer to be marketed internationally. Therefore, it is not surprising that the varnishers of the Viceroyalty of Peru had access to Japanese and Ryukyu lacquerware during the 17th and even 18th centuries [7].

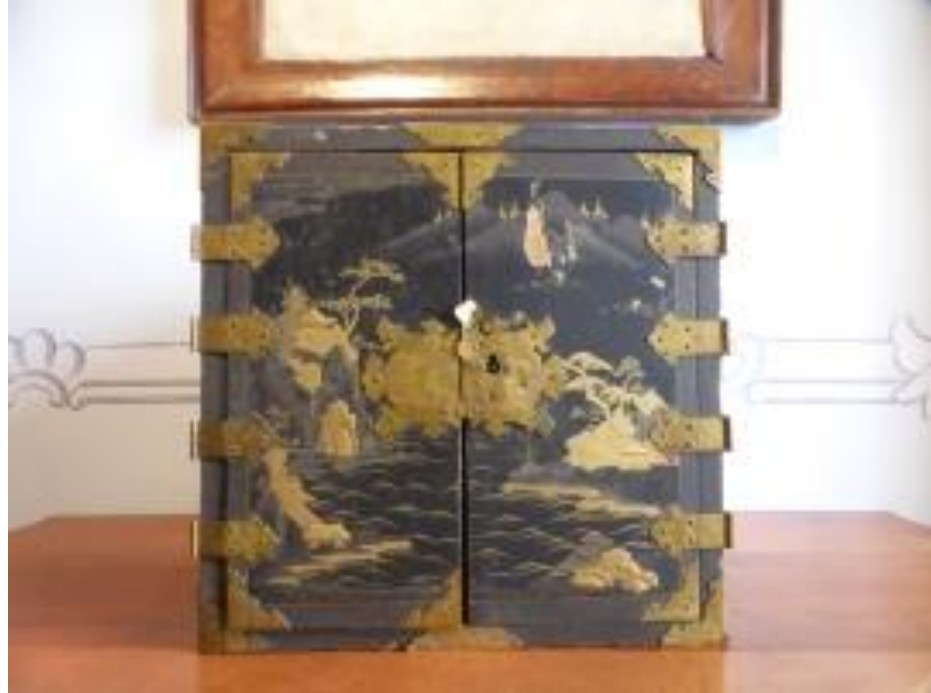

**Figure 12.** Pictorial style lacquer cabinet, Casa de Alfeñique, Puebla, Mexico. Japan, 1640–1700. Photography by the author.

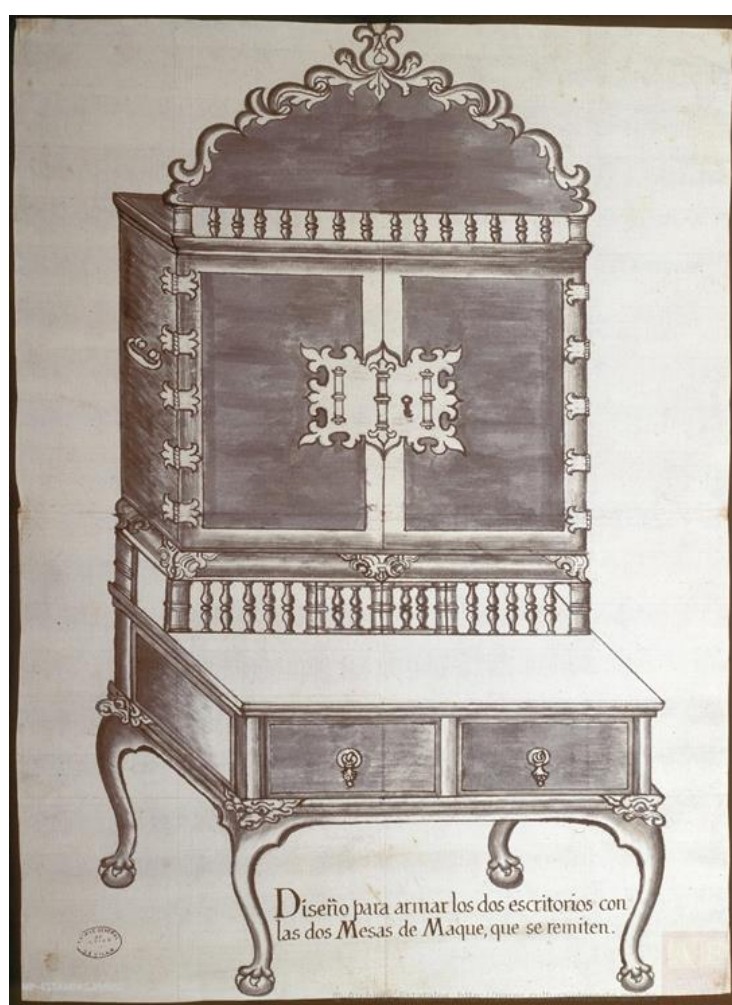

**Figure 13.** Drawing of a Pictorial style lacquer cabinet, made in Manila, ca. 1720. Archivo General de Indias (General Archive of the Indies), Seville, Spain.

## 3. Printed Images of European Origin Applied in *Barniz de Pasto*

A question that has been debated among researchers is the place or places of production of *barniz de Pasto*. Although the practice is currently limited to the southwest of Colombia, around Pasto, López points out that there was noteworthy activity in Quito, likely linked to the workshops operated by the Franciscans. Additionally, the practice of this art in the Central Andean zone has not been ruled out [15].

Based on an iconographic analysis of a *barniz de Pasto* coffer from a private collection (Figure 14), I would like to delve deeper into this issue. Although the work was previously studied [30], the present study attempts to reveal new clues that could hint at its possible production in the Central Andes. The coffer is made with glossy *barniz de Pasto*, without relief. In addition to the application of a rich color range on a black background, the presence of silver leaf between translucent sheets of different colors creates an intense and highly attractive shine. The coffer is densely decorated with various tropical motifs of flora and fauna, in addition to some human figures and fantastical animals.

On the front, there are two human figures with very schematic bodies holding an object in the shape of a heart or shield in their hands. Below them there is an inscription that reads "O ME LO LLEVO/NO SINO IO" ("I TAKE THIS FOR MYSELF/IF NOT, I TAKE IT"). According to the study of emblems widespread in Spain during the 16th and 17th centuries [33], a heart between two men represents trust or agreement, as is shown in the emblems created by Sebastián de Covarrubias Horozco (1610) and Francisco de Zárraga

(1684) [34] (pp. 233–234). In this case, it is likely to symbolize an agreement between two parties, and the text can be interpreted as an expression of trust.

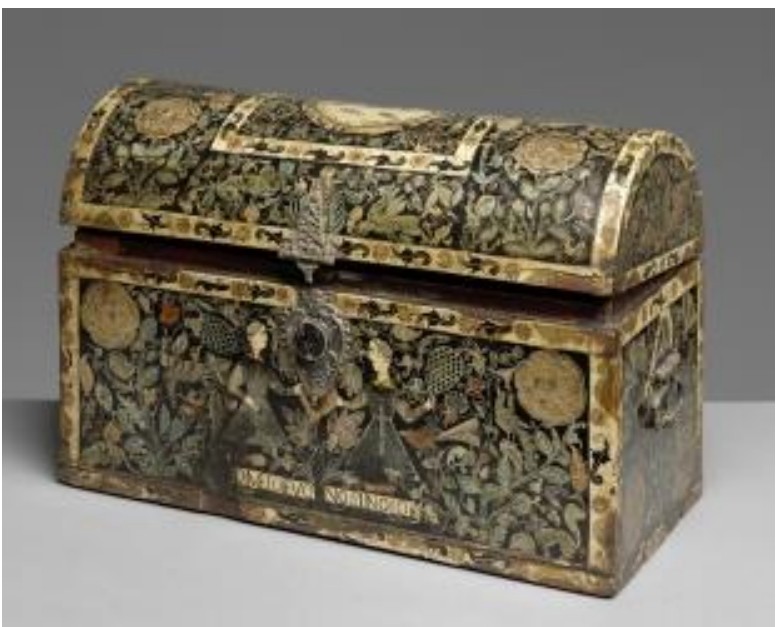

**Figure 14.** *Barniz de Pasto* coffer, private collection. Viceroyalty of Peru, 17–18th century. Photography by Marcos Morilla.

In the center of the rear panel is a crowned phoenix with extended red and gold wings. It rises over a grate with fire and heads skyward (Figure 15). This is the usual iconography of this mythological bird, a symbol of resurrection and eternal life in Christianity. Above the bird the text reads: "EX ME IPSO RENASCOR", which means, "from myself I am reborn". The phrase, put into the mouth of the immortal bird, and its very image constitutes one of the widespread emblems and, as Esteban [35] points out, it was the mark of the Zaragozan publisher Juan de Bonilla, as we can see on the cover of the book *Emblemas morales*, by Juan de Horozco y Covarrubias (1604) [36] (Figure 15).

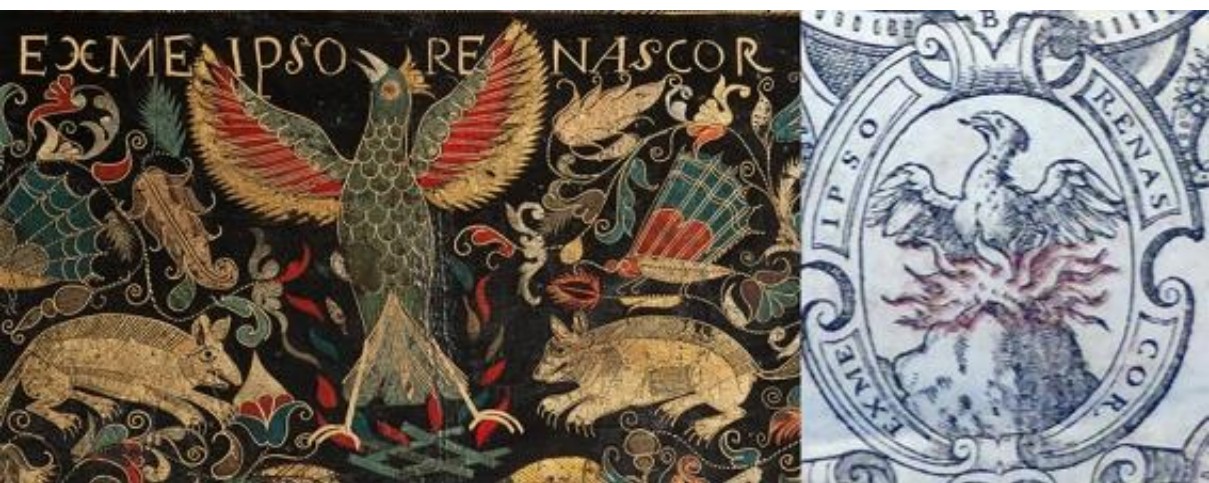

**Figure 15.** Previous coffer detail. Photography by Marcos Morilla (**left**). The cover of the book *Emblemas morales*, by Juan de Horozco y Covarrubias (1604) (**right**).

In the center of the lid, there is a circle in which the aquatic world is represented. Among the waves swim eight fish and a fantastical animal, with a scaly body and a forked tail (Figure 16). It may represent the serpent Amaru, the Quechua deity of water.

Meanwhile, on each of the sides of the coffer, an elongated white serpentine body with a skull, a symbol of death, can be seen. The undulating body serves as an inscribed band, and above it, the Latin phrases "Mortals, be prepared"[5] and "for he who thinks, no value"[6] can be read. Undoubtedly, these are phrases of Christian teaching. Phylacteries with forked ends like these are very frequent in the aforementioned book *Emblemas morales*, by Juan de Horozco y Covarrubias (1604). Additionally, on both side panels of the lid, there is a winged serpent with the head and front legs of a beast, sticking out its red tongue. This animal is also identified as the serpent Amaru (Figure 16), a scaled and winged being with the body of a snake and the head of a llama (*Lama glama*, an animal native to the Andes). As mentioned above, in the Quechua culture, which extends throughout the Central Andes, Amaru is a deity of water, and also of fertility and wisdom. Evidently, the artist used different images of the Spanish emblems taken from a book that was in circulation and combined them with the figure of Amaru from his native culture.

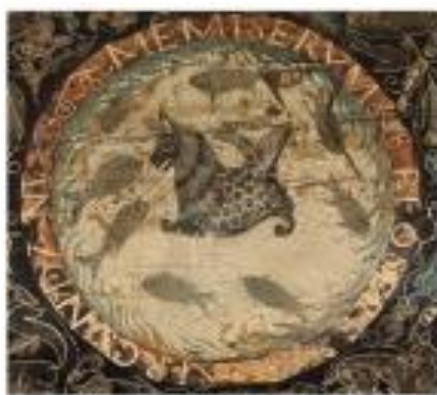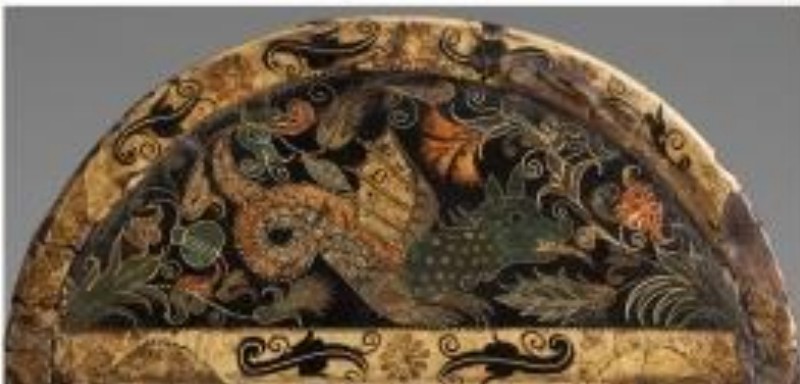

**Figure 16.** Detail from the lid (**left**) and from the side (**right**) of the previous coffer. Photography by Marcos Morilla.

Based on the iconography analyzed here, I intend to propose a possible place of production. Two main areas have been considered to be centers of production during the viceregal period for *barniz de Pasto*: the Pasto–Quito region, where objects of European forms were mainly made with *Elaegia pastoensis*, and the Andean region of Bolivia and Peru, where it is believed that the production of decorated *quero* vases with *Elaegia utilis* originated [6] (p. 107) [11,16]. The coffer's form is linked to the first region. Additionally, the coffer is made of glossy *barniz de Pasto*, a technique only used in the works produced in the workshops of the Pasto–Quito region. However, the presence of Amaru, the deity of the Central Andes, links the work to the second region. Furthermore, the use of a black background and a variegated composition with plant elements, birds, animals, and human beings is reminiscent of a painted coffer from the Museo Casa de Murillo in La Paz, made in Charazani (Bolivia) in the Quechua area [37], in which the same serpent Amaru appears. These aspects make the work very interesting. Although the activities of varnishers in the Central Andes during the viceregal period are largely unknown, this work could have been made in that area.

Additional elements that may corroborate this hypothesis are the appearance of the phoenix among the flames on the back of the coffer and the slogan "EX ME IPSO RENASCOR", which correspond exactly with the mark of the publisher Juan Bonilla (Figure 15). A very similar phenomenon, which is the appropriation of the mark that appears on a very visible page of a European book in a Spanish American painting, can be observed in the Central Andean region, a mural painting of the Church of Carabuco (Bolivia) choir, located next to Lake Titicaca [38], a Quechua cultural area. In the center of the choir, located in the upper part of the entrance to the church, there is a large golden compass and the phrase "Labore et Constantia", which corresponds with the mark of the well-known Antwerp printer Plantin-Motetus, whose religious publications, such as the missal published in 1737,

were spread throughout the Viceroyalty of New Spain. The painting is believed to have been commissioned by the cacique Agustín after the collapse of this part of the church in 1763. Therefore, the impact of an external influence can be observed in both the painting and the previously discussed coffer. A well-known image on a visible page of a European book is appropriated. The golden compass of the Church of Carabuco may indicate that the coffer's origins can be traced back to the Quechua region. Considering the presence of the Quechua deity Amaru on the same coffer, the thesis that the coffer was made in the Central Andes is tenable.

## 4. Conclusions

In the attempt to clarify different aspects of the art of *barniz de Pasto*, Asian influences have recently been highlighted, in addition to those of the Europeans. These influences can be traced back to Japanese and Ryukyu lacquer, as well as to Asian textiles that arrived via Manila galleons to the port of Acapulco, which Peruvian merchants later transported to the Viceroyalty of Peru. Considering the existing commercial traffic and the material similarities of both arts, it is likely that the makers of *barniz de Pasto* came into contact with Japanese export lacquer. I also propose the possibility that the workshops where the *barniz de Pasto* pieces were made were located, apart from in the Pasto–Quito area, in the Central Andes, Quechua territory, based on the iconography of Amaru and the appropriation of a mark of the Spanish printer on a coffer. In this way, I can point out the confluences of different cultures, from Asia and Europe, and their fusion and appropriation in the artistic activity of *barniz de Pasto*.

**Supplementary Materials:** The following supporting information can be downloaded at: https://www.mdpi.com/article/10.3390/heritage6030174/s1, Supplementary material: original text in Spanish.

**Funding:** This research was funded by Agencia Estatal de Investigación of the Spanish Government, reference number PGC2018-097694-B-I00.

**Data Availability Statement:** The author confirms that the data supporting the findings of this study are available within the article.

**Acknowledgments:** I sincerely thank Natasha Vargas (translator), as well as Robert Esposito and Laura Ramirez Polo (reviewers), from Rutgers University for the English version of this article.

**Conflicts of Interest:** The author declares no conflict of interest.

## Notes

[1] The seminar "Las lacas" organized by director Concepción García Saiz on 27 October 2017, and the "Jornada del Barniz de Pasto" organized by the curator of viceregal art Ana Zabala on 27 June 2019, at the Museo de América, Madrid, Spain.

[2] *horror vacui* means the "fear of empty space", or the filling of the entire surface.

[3] The first letter is preserved in the Municipal Archive of Seville (ICAS-SAHP, 3779) and the second in the Vatican Apostolic Library (Borghese 363b).

[4] General Archive of the Indies (Archivo General de Indias), Filipinas, 132, 22–24. Memory of the gift from the Siam's king to our king by his ambassador, sent by Fernando Manuel de Bustillo. Drawing for mounting the two writing desks with two lacquered desks that are shipped.

[5] "mor[ . . . .]s estote [.]arati".

[6] "cogi [.a]ntivi lec si[t]".

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
