# Peer review of "The Art of Barniz de Pasto and Its Appropriation of Other Cultures"

_heritage, doi:10.3390/heritage6030174_

Round 1

Reviewer 1 Report

I do not have comments or suggestions for author/s

Author Response

Thank you very much for your review.

I agreed with the reviewer.

Reviewer 2 Report

inscriptions are transcribed in some instances, not in others.  should be in each case.

I don't understand why phylacteries are cited in an article that otherwise is concerned with Christian iconography.

Why would a borrowing from a book, that might easily have travelled, helped locate this particular object's production? And is this God Amaru, a marine deity, particularly associated with the central Andes?

the photographs are not very clear

typo in n. 41.  Also on p. 12, I believe 41 is meant rather than 51

Is there evidence of other kinds of contact between Quito and the central Andes?

Author Response

Thank you veru much.

I agreed with the reviewer´s comments. I have revised my text according to them.

Reviewer 3 Report

This was a strong contribution to the field. In future research, consider the crossover effects of cross-cultural exchange - it is rarely one-way.

Author Response

(The authors gave the same response as above.)
